# Modulation of Primary Cilia by Alvocidib Inhibition of CILK1

**DOI:** 10.3390/ijms23158121

**Published:** 2022-07-23

**Authors:** Elena X. Wang, Jacob S. Turner, David L. Brautigan, Zheng Fu

**Affiliations:** 1Department of Pharmacology, University of Virginia, Charlottesville, VA 22908, USA; exw7rb@virginia.edu (E.X.W.); jst7ee@virginia.edu (J.S.T.); 2Department of Microbiology, Immunology and Cancer Biology, University of Virginia, Charlottesville, VA 22908, USA; db8g@virginia.edu

**Keywords:** CILK1, ICK, kinase inhibitor, Alvocidib, primary cilia, ciliopathy

## Abstract

The primary cilium provides cell sensory and signaling functions. Cilia structure and function are regulated by ciliogenesis-associated kinase 1 (CILK1). Ciliopathies caused by *CILK1* mutations show longer cilia and abnormal Hedgehog signaling. Our study aimed to identify small molecular inhibitors of CILK1 that would enable pharmacological modulation of primary cilia. A previous screen of a chemical library for interactions with protein kinases revealed that Alvocidib has a picomolar binding affinity for CILK1. In this study, we show that Alvocidib potently inhibits CILK1 (IC_50_ = 20 nM), exhibits selectivity for inhibition of CILK1 over cyclin-dependent kinases 2/4/6 at low nanomolar concentrations, and induces CILK1-dependent cilia elongation. Our results support the use of Alvocidib to potently and selectively inhibit CILK1 to modulate primary cilia.

## 1. Introduction

Cilia and flagella are microscopic structures protruding from the cell surface that provide sensory functions and motility. Acting as cellular antenna, the primary cilium in animal cells senses environmental cues and transduces and integrates extracellular signals [1,2]. Structural integrity of the primary cilium is essential for these sensory and signaling functions. Ciliopathies are a spectrum of human disorders attributed to cilia dysfunction [3,4]. Alterations of primary cilium and ciliary signaling events are implicated in neurological disorders and cancer development [5,6,7]. In neurons, the primary cilium plays a key role in determining the signaling and responsive state and establishing a non-synaptic mechanism for interneuron connectivity [8,9,10]. In cancer cells, numerous oncogenic proteins, such as Smoothened and KRAS, are compartmentalized in the primary cilium [11]. Upon acquisition of drug resistance, cancer cells show increased cilia length and Hedgehog signaling [12]. Thus, targeting the primary cilium has potential to modulate cancer cell response to kinase inhibitors and may offer new opportunities to overcome drug resistance.

One approach to target the primary cilium is by the inhibition of ciliogenesis associated kinase 1 (CILK1). This kinase was formerly named intestinal cell kinase (ICK) after the tissue it was cloned from [13]. CILK1 is highly conserved among eukaryotes and functions as a negative regulator of cilia length [14]. Human *CILK1* mutations that compromise the kinase activity are associated with ciliopathies that manifest in longer cilia and abnormal ciliary signaling [15,16,17]. *CILK1* variants were recently linked to epilepsy [18]. These epileptic mutants lose the ability to restrict cilia length [19]. However, how they disrupt interneuron activities and cause seizures remain unknown. Although knockout and transgenic models have revealed CILK1 functions, a pharmacological inhibitor would be a useful tool to help understand the mechanisms of CILK1 actions in human diseases.

In this study, we identified Alvocidib (flavopiridol), a semi-synthetic flavone related to a natural product extracted from Indian plants, as a potent and relatively selective inhibitor for CILK1 in cells. Although Alvocidib had been widely used as a pan-CDK (cyclin-dependent protein kinase) inhibitor [20], we demonstrate that at low nanomolar concentrations in cells it shows selectivity for the inhibition of CILK1 over CDK2/4/6. We further show that Alvocidib induces a significant, CILK1-dependent increase in cilia length. Our results show the potential of targeting primary cilia with Alvocidib.

## 2. Results

### 2.1. Alvocidib Potently Inhibits Phosphorylation of KIF3A by CILK1 in Cells

Alvocidib was identified as a ligand for CILK1 by screening of a chemical library for binding to kinases (http://www.kinasenet.ca/, accessed on 26 April 2021). The results revealed that CILK1 has the highest binding affinity for Alvocidib (Kd = 690 pM), followed by R547 (Kd = 2.2 nM) and AT7519 (Kd = 8.3 nM). All three compounds are known as ATP-competitive inhibitors of CDKs, which share significant sequence alignment with CILK1 and structural similarities in the catalytic domain. We tested for the inhibition of CILK1 activity in cells by Alvocidib and AT7519. We co-expressed either GST-CILK1 or Flag-CILK1 with HA-KIF3A as a preferred CILK1 substrate in HEK293T cells. We treated cells with increasing concentrations of Alvocidib and assessed the phosphorylation of KIF3A-Thr672 by immunoblotting for total KIF3A and phosphosite pT672 in both cell lysates and anti-HA immunoprecipitates. As shown in Figure 1A, cells treated with 10 nM Alvocidib showed a marked decrease in phospho-KIF3A-Thr672 compared with cells treated with DMSO (as solvent control) or 10 pM Alvocidib (Figure 1A). At 1 µM, Alvocidib eliminated the phosphorylation of KIF3A-Thr672. The sensitivity of CILK1 to inhibition by Alvocidib was about the same for both tagged versions, with some difference noted at the 10 nM dose (Figure 1A). We also compared the effect of Alvocidib vs. AT7519 on the phosphorylation of KIF3A by GST-CILK1 (Figure 1B). While 100 nM Alvocidib appeared to fully inhibit KIF3A phosphorylation, there was only a modest reduction in phospho-KIF3A at a 10-fold higher concentration of 1 µM AT7519 (Figure 1B). The dose-response curves (Figure 1C) indicated that Alvocidib effectively inhibited CILK1 activity in cells, in contrast to AT7519, a much less potent inhibitor.

We subsequently evaluated the effect of Alvocidib on CILK1 activity over time. Alvocidib (100 nM) essentially eliminated KIF3A-Thr672 phosphorylation by either GST-CILK1 or Flag-CILK1 after 15 min treatment, and this inhibition remained complete for at least 2 h post treatment (Figure 1D). Our data show that Alvocidib exerts an acute and sustained inhibitory effect on CILK1 activity in living cells.

Next, we co-transfected HEK293T cells with GST-CILK1 and HA-KIF3A and then treated cells with Alvocidib at a range of concentrations in the low nanomolar range, based on data in Figure 1A,B. We analyzed phospho-KIF3A Thr672 in both cell lysate and anti-HA immunoprecipitate samples. In both analyses we observed the same extent of inhibition of CILK1 phosphorylation of KIF3A (Figure 1E). Plotting the data revealed a 50% inhibition of CILK1 in cells at 20 nM Alvocidib (Figure 1F).

### 2.2. Alvocidib Does Not Inhibit Phosphorylation of the TDY Motif in CILK1

In addition to inhibition by competing for ATP binding, one way to reduce CILK1 activity in cells would be to interfere with its activation by phosphorylation of the TDY motif. Phosphorylation of the TDY motif in the CILK1 activation loop is essential for kinase activation [21]. We extracted GST-CILK1 or Flag-CILK1 proteins from extracts of cells treated with vehicle or Alvocidib by GST pulldown or anti-Flag immunoprecipitation, respectively. We immunoblotted CILK1 proteins with antibodies that recognize total CILK1 or the phospho-TDY motif (Figure 2A,B). Our quantification showed that Alvocidib had little effect on CILK1 phosphorylation in its TDY motif under conditions that fully inhibited kinase activity with KIF3A as substrate (Figure 2C). This result indicates that Alvocidib blocked the activity of the activated, phosphorylated form of CILK1.

### 2.3. Alvocidib at Low Doses Displays Selectivity for CILK1 over CDK in Cells

Alvocidib inhibits CDK1, CDK2, CDK4, CDK6, and CDK9 with IC_50_ values in the 20–100 nM range in biochemical assays [22,23,24]. Thus, it has been promoted and used as a CDK inhibitor in cells. Indeed, in a cell-based assay we found Alvocidib (1 μM) inhibited CDK phosphorylation of retinoblastoma protein (Rb) at Ser807/811. By comparison, Alvocidib effectively inhibited CILK1 activity in cells at much lower concentrations, in the 20–100 nM range (Figure 1E,F). We sought to determine whether Alvocidib in this low nanomolar range would also inhibit CDK activity. We used phosphosite specific immunoblotting for Rb phosphorylation on Ser807/811 (CDK2 sites) and Ser780 (CDK4/CDK6 site). Our results showed that Alvocidib added to cells in the 20–160 nM range did not cause a significant reduction of Rb phosphorylation on these sites (Figure 3). We concluded that low nanomolar doses of Alvocidib were not sufficient to inhibit CDK2/4/6 in living cells. Thus, at these low nanomolar concentrations, Alvocidib exhibits selectivity for the inhibition of CILK1 versus some CDKs in cells. It is worth pointing out that as CDK4/6 phosphorylate the same target residue on Rb, we could not exclude the possibility that the sensitivity of one kinase to the inhibitor may be concealed by the insensitivity of the other kinase.

Alvocidib can arrest the cell cycle by causing not only inhibition of CDKs but also downregulation of cyclin D1 and D3 [25]. However, we did not observe any significant decline in the protein levels of cyclin D1 or D3 in cells treated with low nanomolar concentrations of Alvocidib (Figure 3). Together, these data indicated that at low doses Alvocidib selects CILK1 over cell cycle targets (CDK2/4/6 and cyclin D1/D3) for inhibition in MEF cells.

### 2.4. Alvocidib Modulates Cilia Length via CILK1 Inhibition

CILK1 is a negative regulator of cilia length, and we tested if Alvocidib inhibition of CILK1 would induce cilia elongation. We used mouse embryonic fibroblast (MEF) cell lines that express either wild type (WT) CILK1 or the R272Q mutant form. Our prior work has shown that the R272Q mutation in CILK1 eliminates kinase activity and there is cilia elongation in MEFs expressing this CILK1 inactive mutant [16]. We treated MEF cell lines expressing either WT or the R272Q mutant with Alvocidib (100 nM, IC_90_) or DMSO (solvent control) for 4 h and then fixed and immunostained cells with cilia marker Arl13B and basal body marker γ-tubulin (Figure 4A). We measured cilia length in Z stacks of confocal immunofluorescence images (Figure 4B). In WT MEF cells, Alvocidib treatment induced a 20% increase in cilia length (Figure 4B). However, in cells expressing the CILK1 R272Q mutant, Alvocidib did not induce any statistically significant increase in cilia length (Figure 4B). We thus concluded that Alvocidib stimulates the elongation of primary cilia, and this response is dependent on the inhibition of CILK1.

## 3. Discussion

In this study, we show that Alvocidib offers a new pharmacological tool to modulate primary cilia by the inhibition of CILK1. Cilium elongation as a result of Alvocidib inhibition of CILK1 is consistent with prior observations that genetic deletion or inactivation of *Cilk1* in mice caused abnormal extension of the cilium [15,16] and over-expression of CILK1 in cells decreased cilium length [17,19,26]. In the RCK (*v-ros*
cross-hybridizing kinase) branch of the protein kinome, CILK1 is clustered with MAK (male germ cell-associated kinase) and MOK (MAPK/MAK/MRK-overlapping kinase) [14]. These RCK kinases and their homologs are conserved negative regulators of cilium and flagellum length [15,27,28,29,30,31].

Since MAK is mostly related to CILK1 in sequence and both are negative regulators of cilia length, is it possible that Alvocidib inhibition of MAK also contributed to cilia elongation. CILK1 and MAK are 88% identical in the catalytic domain. It is therefore unsurprising that they share the top three drugs (Alvocidib, R547, and AT7519) in the ranking of drug-target binding affinity (http://www.kinasenet.ca/, accessed on 26 April 2021). However, Alvocidib reportedly has a 40-fold higher binding affinity for CILK1 (Kd = 0.69 nM) than for MAK (Kd = 28 nM), suggesting that Alvocidib has a strong preference for CILK1 over MAK. Therefore, we postulated that Alvocidib at the low nanomolar range can selectively inhibit CILK1 but not MAK in cells, which remains to be determined. It is worth pointing out that an insignificant elongation of primary cilium by 100 nM Alvocidib in Cilk1 mutant cells suggests that MAK is likely not a primary target by Alvocidib at low nanomolar concentrations in cells.

Alvocidib inhibited CILK1 activity against KIF3A in cells without affecting TDY motif phosphorylation in CILK1 itself. This indicated Alvocidib inhibition of CILK1 kinase but not its activation. Alvocidib has been shown to inhibit CDKs by competing with ATP for the active site [24]. It is therefore likely that Alvocidib targets the ATP binding site in CILK1 for inhibition. Which residues in the ATP binding pocket of CILK1, compared with CDKs and MAK, confer a stronger binding affinity to Alvocidib merits further investigation.

In this study, we used KIF3A as a substrate to assess the effect of Alvocidib on CILK1 catalytic activity in cells. Alvocidib showed similar effects on CILK1 phosphorylation of another CILK1 substrate, GSK3β (Appendix A). Both KIF3A and GSK3β have been linked to intraflagellar transport and cilia length control [32,33]. However, our prior study showed that deficiency in CILK1 phosphorylation of KIF3A alone is insufficient to account for the cilia phenotype caused by CILK1 inactivation [34]. Therefore, it remains a mystery as to which CILK1 substrate(s) mediate the effect of Alvocidib in the regulation of intraflagellar transport and cilia length.

## 4. Materials and Methods

### 4.1. Reagents and Plasmids

Alvocidib/flavopiridol (S2679) and AT7519 (S1524) were from Selleckchem (Houston, TX, USA). Glutathione Sepharose 4B (17-0756-01) and Gammabind Plus Sepharose beads (17-0886-01) were from GE Healthcare (Chicago, IL, USA). Anti-FLAG tag affinity beads (ab270704) were from Abcam (Cambridge, MA, USA). pEBG-GST-CILK1/ICK encoding GST-CILK1/ICK was described in [13,21]. pCMV6-Myc-Flag-CILK1 encoding CILK1-Myc-Flag (RC213609) was from Origene (Rockville, MD, USA). pCIG-HA-KIF3A encoding HA-KIF3A was described in [35].

### 4.2. Antibodies

KIF3A phospho-Thr672 rabbit polyclonal antibody was generated at GenScript (Piscataway, NJ, USA) and described in [35]. GST-tag (B-14) mouse monoclonal (sc-138) and HA-tag (12CA5) mouse monoclonal (sc-57592) antibodies were from Santa Cruz Biotechnology (Dallas, TX, USA). KIF3A (D7G3) rabbit monoclonal (#8507) and HA-tag (C29F4) rabbit monoclonal (#3724) antibodies were from Cell Signaling Technology (Danvers, MA, USA). Arl13B rabbit polyclonal antibody (17711-1-AP) and Gamma-tubulin mouse monoclonal antibody (66320-1-Ig) were from Proteintech (Rosemont, IL, USA). Goat anti-rabbit IgG (Alexa Fluor 488) antibody (ab150081) and goat anti-mouse IgG (Alexa Fluor 594) antibody (ab150120) were from Abcam (Cambridge, MA, USA).

### 4.3. Cell Culture and Transfection

Mouse embryonic fibroblast (MEF) cells were isolated from Cilk1 wild type and R272Q mutant embryos (E14.5-E15.5) [16]. HEK293T and MEF cells were maintained at 37 °C and 5% CO_2_ in Dulbecco’s modified Eagle’s medium (DMEM) supplemented with 4.5 g/L glucose, 10% fetal bovine serum, and penicillin-streptomycin. HEK293T cells were transfected using a calcium phosphate protocol as described in [36].

### 4.4. GST Pull-Down, Immunoprecipitation, and Immunoblotting

Forty-eight hours after transfection, cells were lysed in lysis buffer (50 mM Tris-HCl, pH 7.4, 150 mM NaCl, 1% NP-40, 2 mM EGTA, complete protease inhibitors (Roche), 10 mM sodium orthovanadate, 5 mM sodium fluoride, 10 mM sodium pyrophosphate, 10 mM β–glycerophosphate, and 1 µM microcystin LR). Cell lysate was cleared by centrifugation. GST-CILK1 proteins were pulled down from cell lysate using Glutathione Sepharose 4B beads (GE Healthcare) following the manufacturer’s instructions. HA-KIF3A proteins were immunoprecipitated from cell lysate using HA antibody and captured on GammaBind Sepharose beads (GE Healthcare).

Cell extracts or Sepharose beads were boiled for 5 min in an equal volume of 2X Laemmli sample buffer (120 mM Tris-HCl, pH 6.8, 4% SDS, 20% glycerol, 10% β-mercaptoethanol, 0.02% bromophenol blue) and loaded on an SDS-gel. Samples were transferred to a PVDF membrane and blocked for one hour in 5% dry milk before primary antibody incubation in TBS containing 0.1% Tween-20 and 5% bovine serum albumin (BSA) for 90 min at RT or overnight at 4°C. This was followed by extensive rinses and one-hour incubation with horseradish peroxidase (HRP)-conjugated secondary antibody. Chemiluminescence signals were developed using Millipore Immobilon ECL reagents from EMD Millipore (Burlington, MA, USA).

### 4.5. Immunofluorescence

MEF cells grown on gelatin-coated coverslips were fixed by 4% paraformaldehyde (PFA) in PBS, rinsed in PBS, and then permeabilized by 0.2% Triton X-100 in PBS. After one hour in blocking buffer (3% goat serum, 0.2% Triton X-100 in PBS), MEF cells on cover slips were incubated with primary antibodies at 4 °C overnight followed by rinses in PBS and one hour incubation with Alexa Fluor-conjugated secondary antibodies. After extensive rinses, slides were mounted in antifade reagent containing DAPI (4’,6-diamidino-2-phenylindole) for imaging via a confocal Laser Scanning Microscopy 700 from ZEISS (Chester, VA, USA).

### 4.6. Cilia Length Measurement

The Zen 2009 program was used with a confocal Laser Scanning Microscope 700 from ZEISS (Chester, VA, USA) to collect z stacks at 0.5 μm intervals to incorporate the full axoneme based on cilia marker Arl13b staining. All cilia were then measured in Fiji/ImageJ (version 1.52t) (created by Wayne Rasband, National Institute of Mental Health, Bethesda, MD, USA) via a standardized method based on the Pythagorean Theorem in which cilia length was based on the equation L2 = z2 + c2, in which “c” is the longest flat length measured of the z slices and “z” is the number of z slices in which the measured cilia were present multiplied by the z stack interval (0.5 μm).

### 4.7. Statistical Analysis

Quantified experimental data were analyzed by the Student’s *t*-test. Data were reported as mean ± standard deviation (SD). *p*-values less than 0.05 were considered as significant.

## 5. Conclusions

We have demonstrated here that Alvocidib at low doses can modulate primary cilia via potent and selective inhibition of CILK1. We believe that Alvocidib offers a useful new tool compound to study the role of the primary cilium in cell signaling related to neural circuit function and cancer chemoresistance.

## Figures and Tables

**Figure 1 ijms-23-08121-f001:**
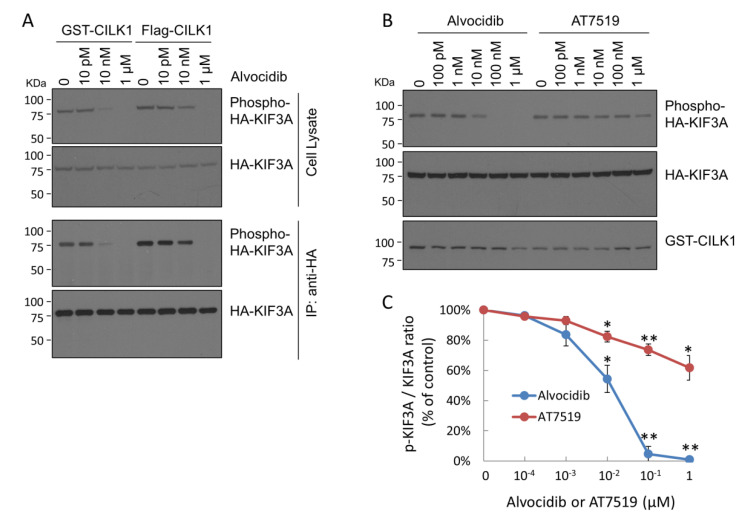
Alvocidib effect on CILK1 phosphorylation of KIF3A in cells. (**A**) GST-CILK1 or Flag-CILK1 was co-transfected with HA-KIF3A (substrate) into HEK293T cells. Forty-eight hours after transfection, cells were treated with increasing concentrations of Alvocidib in DMSO for 15 min before lysis. HA-KIF3A was immunoprecipitated from cell extracts with mouse monoclonal HA-tag antibody (12CA5). Both cell extracts and anti-HA immunoprecipitates were Western blotted for total and phospho-T672 KIF3A. (**B**) HEK293T cells co-transfected with GST-CILK1 and HA-KIF3A were treated with various doses of Alvocidib or AT7519. Cell extracts were Western blotted for total and phospho-T672 KIF3A and GST-CILK1. (**C**) The phospho-T672 versus total KIF3A ratio was plotted against the concentrations of drugs. Shown is average ± SD, n = 3 independent experiments; * *p* < 0.05; ** *p* < 0.01. (**D**) GST-CILK1 or Flag-CILK1 was co-transfected with HA-KIF3A into HEK293T cells. Cells were treated with 100 nM Alvocidib in a time course. Cell extracts were Western blotted for total and phospho-T672 KIF3A. From cell extracts, GST-CILK1 and Flag-CILK1 proteins were recovered by glutathione agarose beads and anti-FLAG M2 agarose beads, respectively, and then Western blotted by a CILK1 antibody. (**E**) HEK293T cells co-transfected with GST-CILK1 and HA-KIF3A were treated with Alvocidib in the nanomolar range for 15 min before lysis, protein extraction, and immunoprecipitation. Both cell extracts and anti-HA immunoprecipitates were Western blotted for total and phospho-T672 KIF3A. (**F**) Phospho-T672 of HA-KIF3A in anti-HA immunoprecipitates was plotted against various concentrations of Alvocidib. Shown is average ± SD, n = 3 independent experiments; ** *p* < 0.01.

**Figure 2 ijms-23-08121-f002:**
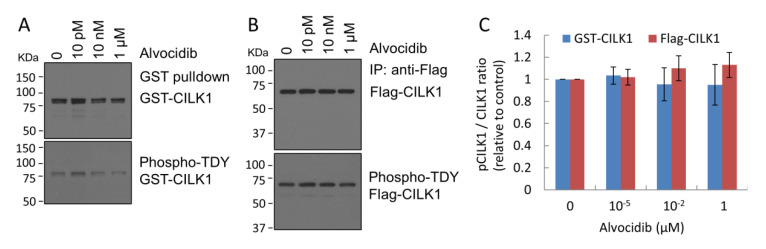
Alvocidib effect on TDY phosphorylation in CILK1. (**A**,**B**) HEK293T cells transfected with either GST-CILK1 or Flag-CILK1 were treated with Alvocidib or DMSO control. After GST pulldown or anti-Flag immunoprecipitation, total and phospho-TDY signals of CILK1 were detected by Western blotting with antibodies recognizing CILK1 and phospho-TDY motif, respectively. (**C**) The phospho-TDY and total CILK1 ratio relative to the DMSO control against different concentrations of Alvocidib is shown as average ± SD, n = 3; Student *t*-test, not significant.

**Figure 3 ijms-23-08121-f003:**
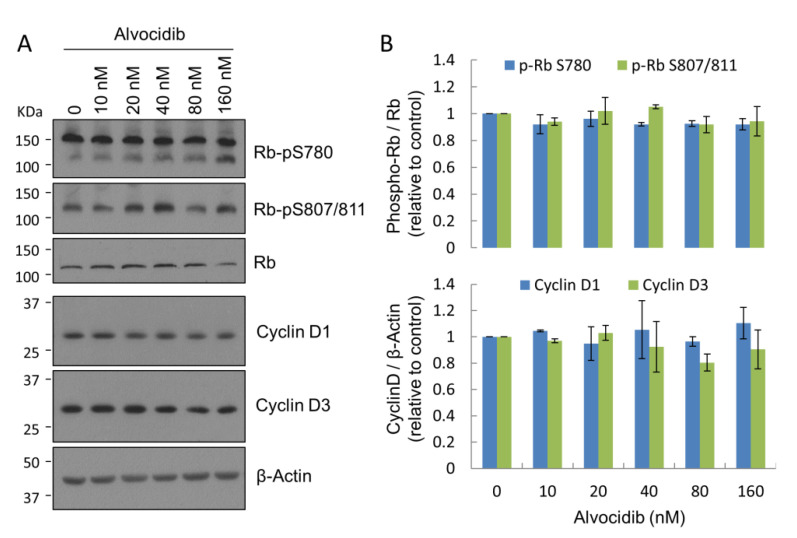
(**A**) Alvocidib effect on Rb phosphorylation and protein levels of cyclin D1/D3. HEK293T cells treated with DMSO or increasing concentrations of Alvocidib. Equal amounts of total proteins from cell lysates were Western blotted for total Rb, phospho-Rb, and cyclin D1 and D3 signals. (**B**) Phospho-Rb signals were normalized against total Rb and Cyclin D1 and D3 signals were normalized against β-Actin. Shown is average ± SD, n = 3; Student *t*-test, not significant.

**Figure 4 ijms-23-08121-f004:**
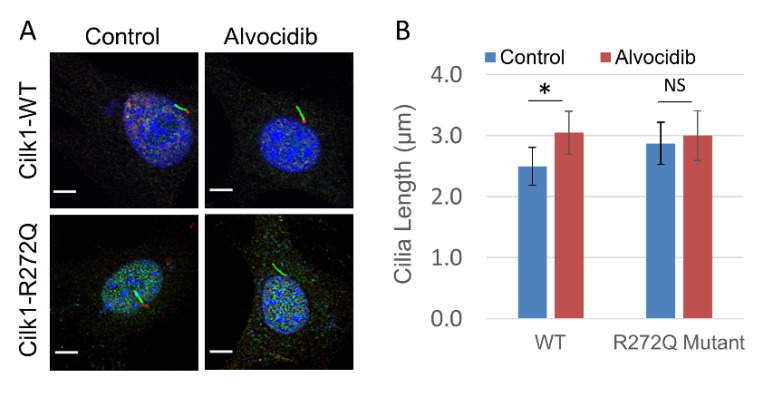
Alvocidib effect on cilia length in wild type and Cilk1 R272Q mutant cells. (**A**) Wild type (WT) or the R272Q mutant mouse embryonic fibroblasts (MEFs) were treated with either DMSO control or Alvocidib (100 nM) for 4 h, fixed, and then co-immunostained for the cilium body (anti-Arl13B, green), the cilium base (anti-γ-tubulin, red), and the nucleus (DAPI, blue). Scale bar, 3 µm. (**B**) Quantification of cilia length (mean ± SD) in WT and R272Q mutant MEF cells that were treated with either DMSO control or Alvocidib. WT/control (n = 96 cilia); WT/Alvocidib (n = 75 cilia); R272Q/control (n = 53 cilia); R272Q/Alvocidib (n = 48 cilia). * *p* < 0.05; NS, not significant.

## Data Availability

The data presented in this study are available in the article.

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
