# Peer review of "Modulation of Primary Cilia by Alvocidib Inhibition of CILK1"

_ijms, 2022, doi:10.3390/ijms23158121_

Round 1

Reviewer 1 Report

In this manuscript, Wang and colleagues characterized the inhibitory action of Alvocidib on Cilk1. They use KIF3A phosphorylation as the readout and show that Alvocidib inhibits CILK1 at the nanomolar level. They further showed that CDK mediated RB phosphorylation is not inhibited by Alvocidib at the same nanomolar concentration. Finally, they showed that Alvocidib treatment leads to cilia elongation in cells expressing wild type, but not a mutant form of CILK1, suggesting that it promote cilia elongation through a CILK1-dependent mechanism. The manuscript is clearly written, and the data are of decent quality. The identification of a specific CILK inhibitor and cilia length regulating reagent is significant to the cilia field.

Specific Comments

1.     All experiments reported should be repeated independently for at least three times. Please explicitly state this fact (in the method section or figure legends) if true. Otherwise please repeat the experiments as needed and add stand deviation to the charts.

2.     I have two questions on the CDK experiments. First, the author mentioned that it was known that Alvocidib inhibited five CDKs, but the experiments only tested three (CDK2/4/6). Therefore, the authors should be more cautious in wording, and avoid use the general term CDK as they do not have data on CDK1 and CDK9. Second, as CDK4/6 phosphorylate the same target residue on RB, sensitivity of one kinase to the inhibitor may be concealed by the insensitivity of the other kinase. The authors should comment on their relative sensitivity in the manuscript.

3.     A chart(s) can be added to Figure 3 to show the quantification result as the authors did for Figs 1 and 2.

Author Response

Reviewer #1

Specific Comments

  1. All experiments reported should be repeated independently for at least three times. Please explicitly state this fact (in the method section or figure legends) if true. Otherwise please repeat the experiments as needed and add standard deviation to the charts.

We have added standard deviation bars to the graphs and indicated the number of replicates and statistical significance in figure legends (Fig. 1C; Fig. 2C; Fig. 3B). 

  1. I have two questions on the CDK experiments. First, the author mentioned that it was known that Alvocidib inhibited five CDKs, but the experiments only tested three (CDK2/4/6). Therefore, the authors should be more cautious in wording, and avoid use the general term CDK as they do not have data on CDK1 and CDK9. Second, as CDK4/6 phosphorylate the same target residue on RB, sensitivity of one kinase to the inhibitor may be concealed by the insensitivity of the other kinase. The authors should comment on their relative sensitivity in the manuscript.

We agree with the reviewer that we must avoid the use of general term CDK and be more specific in our wording. We have revised accordingly by using CDK2/4/6 instead of CDK or CDKs in the text (page 1, line 15-16; page 2, line 49; page 4, line 137). We also added a comment (page 4, lines 138-140) to indicate that as CDK4/6 phosphorylate the same target residues on Rb, we could not exclude the possibility that sensitivity of one kinase to the inhibitor may be concealed by the insensitivity of the other kinase.   

  1. A chart(s) can be added to Figure 3 to show the quantification result as the authors did for Figs 1 and 2.

We have added a chart (3B) to Figure 3 to show the quantification results.

Reviewer 2 Report

I thought this paper was very important for future research on the role of primary cilia, as it shows that alvocidib is a selective inhibitor of CILK1. I would like to make two minor comments.

Minor comments

#1 For Results 2-4, you state that mice with the R272Q mutation have loss of CILK1 activity. Could you add a little more explanation on the interpretation of alvocidib treatment for this mutant. I think it means that the loss of CILK1 activity did not inhibit the prolongation of cilia, but it seems a little confusing to me.

#2 In regards to the 200th line. Can you show the effect of alvocidib on GSK3B, even if it is just a supplement?

Author Response

Review #2

Minor comments

  1. For Results 2-4, you state that mice with the R272Q mutation have loss of CILK1 activity. Could you add a little more explanation on the interpretation of alvocidib treatment for this mutant? I think it means that the loss of CILK1 activity did not inhibit the prolongation of cilia, but it seems a little confusing to me.

The R272Q mutant version of CILK1 is inactive as a negative regulator of cilia length, thus there is elongation of primacy cilia in R272Q mutant MEF cells. Alvocidib treatment could not further elongate primary cilia in these R272Q mutant cells, thus demonstrating that the Alvocidib effect on cilia length is CILK1-dependent. We have revised our explanation in the text to make it clearer (section 2.4, page 5).  

  1. In regard to the 200th line. Can you show the effect of alvocidib on GSK3B, even if it is just a supplement?

Yes, we have added a supplemental Figure to show the effect of Alvocidib on GSK3beta.